# Topological RANSAC for instance verification and retrieval without fine-tuning

**Guoyuan An[1], Juhyung Seon[1], InKyu An[1,4], Yuchi Huo[2,3], and Sung-Eui Yoon[1]**

[1]School of Computing, KAIST
[2] State Key Lab of CAD&CG, Zhejiang University
[3]Zhejiang Lab
[4]ETRI, Electronics and Telecommunications Research Institute

## Abstract

This paper presents an innovative visual reasoning approach to enhancing instance verification and retrieval, particularly in situations where a fine-tuning set is unavailable. The widely-used SPatial verification (SP) method, despite its efficacy, relies on a spatial model and the hypothesis-testing strategy for instance recognition, leading to inherent limitations, including the assumption of planar structures and neglect of topological relations among features. To address these shortcomings, we introduce a pioneering technique that replaces the spatial model with a topological one within the RANSAC process. We propose bio-inspired saccade and fovea functions to verify the topological consistency among features, effectively circumventing the issues associated with SP's spatial model. Our experimental results demonstrate that our method significantly outperforms SP, achieving state-of-the-art performance in non-fine-tuning retrieval. Furthermore, our approach can enhance performance when used in conjunction with fine-tuned features. Importantly, our method retains high explainability and is lightweight, offering a practical and adaptable solution for a variety of real-world applications. Our code can be found through this link.

## 1   Introduction and related work

Content-based image retrieval, a fundamental and long-standing challenge, has seen significant advancements due to the rise of deep learning [29, 45, 7, 24]. These data-driven approaches, although successful, often rely heavily on fine-tuning with data from the same domain [24, 46]. However, acquiring such a fine-tuning set can be impractical or costly, particularly in open-world or private scenarios. Moreover, these methods often fall short in providing explainability, a critical factor for real-world applications where search results are integral to decision-making.

In situations where a fine-tuning set is absent, the esteemed SPatial verification (SP) [26] method has demonstrated the highest accuracy in verifying an image pair [28]. However, its dependency on a spatial model for recognition exposes vulnerabilities, particularly to viewpoint changes in 3D objects and neglect of topological relations among key points (Sect. 2.1). To address these issues, we pioneer the replacement of the spatial model with a topological one during the RANdom SAmple Consensus (RANSAC) process. We introduce Homeomorphism Region (Def. 2.1), which circumvents the mentioned problems, providing a size metric superior to the original inlier counts used in SP. Drawing inspiration from human observation, we propose innovative saccade and fovea functions to validate topological relations among keypoints within RANSAC [12] iterations (Sect. 3).

Numerous studies have explored SP [2, 31, 21, 36], but none have made such a direct modification to the spatial model in RANSAC, possibly because earlier benchmarks presented the issues of the spatial model as less severe [28, 26, 27]. However, with the introduction of more challenging benchmarks like ROxford and RParis [28], these problems have become increasingly critical. While some recent

37th Conference on Neural Information Processing Systems (NeurIPS 2023).

deep-learning approaches have explored feature relations, their effectiveness diminishes without a fine-tuning set, and they often lack explainability [45, 40]. In contrast, by adopting the hypothesis-testing strategy with topological rules, our method outperforms even large-scale pre-trained methods in non-fine-tuning scenarios, maintaining high explainability and lightweight nature. Moreover, it can further enhance performance when used alongside fine-tuned features. These results show our novel attempt at topological-based RANSAC is a success.

Our contributions are summarized as follows:

- We propose a novel approach to directly modify the underlying common sense in hypothesis testing. Our method is pioneering in adopting topological common sense, diverging from traditional spatial approaches. This shift paves the way for numerous future research opportunities.
- Drawing inspiration from the human observation process, we introduce innovative saccade and fovea functions to verify topological relations among keypoints
- Our method has demonstrated a significant improvement in accuracy by adopting the hypothesis-and-test strategy with topological rules. Our method surpasses all other methods without fine-tuning. Moreover, our approach can be combined with fine-tuned features to achieve even better performance.
- Our method offers a highly explainable and lightweight solution, which is crucial for real-world applications.

## 2 Motivation and overview

### 2.1 Spatial verification

RANdom SAmple Consensus (RANSAC) is an effective method for handling noisy data [12]. It involves using a hypothesis and test strategy to optimize model parameters to accurately describe inliers iteratively. The model selected is crucial for achieving high inference accuracy. For instance recognition, Spatial verification (SP) [26] employs a spatial model to match features in two sets of points, $P_1$ and $P_2$. It finds applications in a variety of contexts, such as reranking in image search [7], facilitating loop closure detection [20], and serving as a preliminary step in 3D reconstruction [35]. By using RANSAC, SP calculates a transformation matrix $M$ to account for the spatial configuration change between different views. The similarity between the two images can then be determined using the following equation:

$$D_s(P_1, P_2|M) = \sum_{p_1^i \in P_1} [\![\|\mathbf{p}_1^i M - \mathbf{p}_2^i\| < \epsilon]\!], \tag{1}$$

where $\mathbf{p}_1^i$ and $\mathbf{p}_2^i$ represent the locations of $p_1^i \in P_1$ and $p_2^i \in P_2$, respectively. $p_2^i$ is the feature in $P_2$ that is most similar to $p_1^i$. The threshold for the location disparity is denoted by $\epsilon$.

The spatial model is intrinsically intertwined with RANSAC in the field of computer vision [9, 8], and it is particularly well-suited for tasks such as pose estimation where achieving an accurate homography or fundamental matrix is the primary objective [38, 23]. However, despite the wealth of research surrounding the use of RANSAC for pose estimation, its integration within rigid body recognition remains under-discussed in the existing literature. This paper endeavors to address this gap, offering insights into why a spatial model may not be ideally suited for recognition tasks.

SP has been the subject of numerous studies [2, 31, 21, 36]. However, none of the existing works have sought to modify the fundamental spatial model. On the other hand, while near-perfect results have been reported on the original Oxford and Paris benchmarks [26, 27], newer challenges such as complex positives and confusing distractors have recently emerged [28]. To address these challenges, contemporary research often bypasses the RANSAC process, focusing instead on enhancing performance through fine-tuning [29, 7, 24, 40]. Nevertheless, this strategy may not be feasible in private or dynamic scenarios where fine-tuning datasets are unavailable. As discussed in Section 5, tasks involving non-fine-tuning recognition are of significant importance. To the best of our knowledge, in the absence of a fine-tuning set, RANSAC-based SP still stands as the most effective method [28] in ROxford and RParis. This paper aims to delve into the often-overlooked aspect of enhancing the RANSAC process within rigid instance recognition tasks.

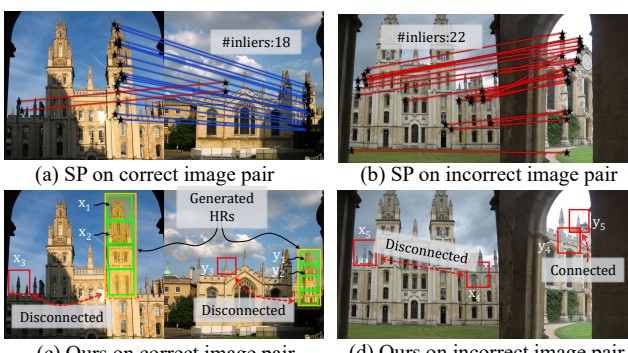

(a) SP on correct image pair      (b) SP on incorrect image pair

(c) Ours on correct image pair      (d) Ours on incorrect image pair

Figure 1: This figure presents our novel adaptation of the SPatial model (SP) [26] into the ToPological model (TP), marking the first such transformation in hypothesis-testing. Sub-figures (a) and (b) highlight an instance where SP incorrectly assigns more inliers to an erroneous image pair (22 inliers) compared to the correct pair (18 inliers). Conversely, sub-figures (c) and (d) demonstrate our TP model accurately discerns the right image pair and dismisses the wrong pair by iteratively refining the Homeomorphism Regions (HRs) based on patch connections.

In our study, we identified two significant issues when spatial model is deployed for recognition. Firstly, SP assumes that two images depicting the same plane from different perspectives can be matched using an affine transformation—an assumption that may not always hold true. Real-world objects are often three-dimensional, not planar, which means the 3 by 3 affine matrix employed in SP may not accurately represent their variations. Secondly, SP's sparse inlier count of the predicted homography matrix neglects the topological relationships among features and the relevance of other image regions. This issue is evident in Fig. 1, which presents SP results for both a correct and an incorrect image pair. Notably, the incorrect image pair generates more inliers than the correct pair, an outcome that starkly illustrates the limitations of using inlier count as the sole measure of accuracy in image recognition tasks. It is important to note that the two issues we've identified with SP primarily originate from the spatial model itself rather than from the hypothesis-testing strategy. This realization motivates us to modify the spatial model while retaining the hypothesis-testing approach.

## 2.2 Topological verification

In contrast to existing works that rely on fine-tuning to enhance performance [40, 7, 25], our study pioneers the use of a topological model in place of the spatial model within RANSAC. Importantly, this approach remains viable even when fine-tuning is not feasible. We introduce the concept of a Homeomorphism Region (HR) to address two primary issues associated with the spatial model. Its size is a new metric for RANSAC iteration and image pair similarity. The formal definition of HRs is in Definition 2.1.

**Definition 2.1 (Homeomorphism region)** *Let $r$ be a small image patch of image $I$, $r_1^n$ and $r_2^n$ be the corresponding patches in $I_1$ and $I_2$, $R_1 = \{r_1^1, ..., r_1^n\}$ and $R_2 = \{r_2^1, ..., r_2^n\}$ be the families of patches of images $I_1$ and $I_2$. $R_1$ and $R_2$ are Homeomorphism regions (HRs), if they satisfy the following conditions:*

*1. Local consistency: $r_1^n \in R_1$ and $r_2^n \in R_2$ are identified as the same patches based on their local descriptors.* [Two corresponding patches should be similar; they are supposed to depict the same region of an object.]

*2. Topological consistency: $\forall (r_1^n, r_1^m, r_2^n, r_2^m)$, if $r_1^n$ and $r_1^m$ overlap each other, $r_2^n$ and $r_2^m$ should also overlap each other, and vice versa.* [Parts of an object that are connected should remain so, even when the viewpoint changes.]

*3. Connectivity: $\forall r_1^n \in R_1, \exists r_1^m \cap r_1^n! = \emptyset, r_1^m \in R_1$* [We do not consider similar but isolated patches. Note that wrong similar patches $x_3$ and $y_3$ in Fig. 1 are excluded from the HR in this way.]

Our novel topological perspective effectively circumvents the aforementioned problems, yielding more robust results than its spatial counterpart. In contrast to SP's planar assumption, our approach posits that parts of an object that are connected should remain so, even when the viewpoint changes.

This assumption holds true for 3D objects. Furthermore, our topological approach considers the topological relationships between patches and leverages more information in an image pair. For example, incorrect SP matches can occur when there are numerous repeated patterns, even if these matched pairs aren't topologically consistent with other features, as shown in Fig. 1 (b). Our model, however, correctly identifies these as different structures. As demonstrated in Fig. 1 (c) and (d), our method successfully discriminates between correct and incorrect image pairs, highlighting its potential in enhancing image recognition tasks.

# 3 Method detail

To find HRs, we are inspired by our brain's mechanism of comparing two first-seen objects. Our eye mostly captures low-resolution visual information except for a tiny patch called the fovea. The fovea only observes a very small area and the entire scene illusion is created by stitching several glimpses of the fovea during eye movements called saccade [17, 14]. Inspired by this process, we design our saccade and fovea functions to detect the HRs that satisfied the Def. 2.1.

Algorithm 1 shows the overall pipeline of our method. Traditionally, RANSAC samples a subset and computes a hypothesis based on this selection. We modify this approach by using each ratio test matching result as an individual hypothesis. Therefore, the size of our sample subset is effectively reduced to one, with each iteration's hypothesis being the translation, as opposed to the affine matrix. Instead of counting inliers based on the spatial constraint, we detect HRs for each hypothesis, the size of which is then used as the new iteration metric. We recognize that the sampling strategy and convergence proof are the focal points of ongoing RANSAC research. However, as the first study to adopt a topological perspective, this paper primarily seeks to demonstrate the feasibility and advantages of this paradigm shift.

---

**Algorithm 1** Topological inference

**Input:** HypothesisSet: $\{(r_1^1, r_2^1),(r_1^2, r_2^2),...\}$.
**Output:** : Regions: list of HRs
1: Regions $\leftarrow$ []
2: **for** $(r_1^i, r_2^i)$ in HypothesisSet **do**
3:    $\pi \leftarrow$ []
4:    $\pi' \leftarrow [(r_1^i, r_2^i)]$
5:    **while** $\pi\prime$ is not empty **do**
6:      $(r_1, r_2)$= pop($\pi\prime$)
7:      $\hat{r}_1, \hat{r}_2, verified$=F($r_1, r_2$)
8:      **if** $verified = True$ **then**
9:        Add $(\hat{r}_1, \hat{r}_2)$ to $\pi$
10:        $\pi'$=S($\pi$)
11:      **end if**
12:    **end while**
13:    add $\pi$ to Regions.
14: **end for**

---

## 3.1 Topological consistency and connection

To verify the connection and topological consistency, we progressively verify and select the patches one by one. It can be expressed as follows:

$$\pi'_t = S(\pi_t), \pi_{t+1} = F(\pi'_t), \tag{2}$$

where $S$ is the saccade function to guide the observation place, $F$ is the fovea function to verify two patches, $\pi_t$ records the verified patches at $t$-th iteration, $\pi_0$ is the given hypothesis about translation, and $\pi'_t$ records the candidate patches generated by $S$. $\pi_t$ and $\pi'_t$ are shown using green and purple boxes in Fig. 2. After finishing this verification progress, the final verified region $\pi_T$ will be treated as the HR defined in Section 3.1. The saccade function $S$ makes the new candidate patch intersect with one and only one verified patch in $\pi_t$, as shown in Fig. 2. In this way, a series of verified patches are connected with each other and also satisfy the topological consistency.

Under the assumption that two images are depicting the same object with some transformations (zoom, foreshortening, vertical shearing), the saccade function $S$ predicts the area around the verified patch pair will be similar. For example, $\hat{r}_1$ and $\hat{r}_2$ in Fig. 2 are two verified patches, and saccade function $S$ expects the patches around them are also similar. The correspondence patch location and size will be refined by $F$, as shown in the next section. Similar to [26], we do not consider in-plane image rotations because of the fact that images are usually displayed with the correct orientation.

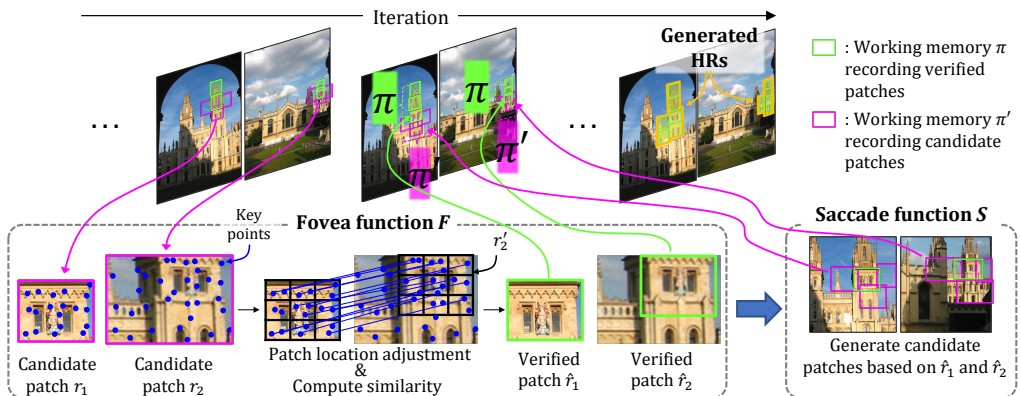

Figure 2: The process of finding HRs for each hypothesis. This is an iterative method. Fovea function F verifies the similarity between two candidate patches at each iteration, and saccade function S updates the candidate patches.

## 3.2 Local consistency

The fovea function $F$ concentrates on two patch areas and verifies if the query patch $r_1$ and test patch $r_2$ are similar based on the local feature sets $P_1$ and $P_2$ of images $I_1$ and $I_2$.

**Feature matching.** The first step for verification is to match the key points in two images correctly. Traditional geometric verification directly searches the nearest neighbor of a keypoint in $I_1$ from keypoints in $I_2$ based on their descriptors. However, we observe this approach may have many wrong matching points in practice due to the repeated patterns in an image, as shown in Fig. 1. To address this problem, we restrict the search region when implementing the fovea function. That is, for each key point in patch $r_1$, we find its nearest neighbor among keypoints in patch $r_2$ instead of $I_2$. We find that this approach increases the matching performance.

**Patch location adjustment.** We need to adjust the location and size of the candidate patches here. Because $r_1$ and $r_2$ are candidate patches generated by saccade function $S$ only based on the verified region $\pi$, their contents may not be exactly same. As shown in Fig. 2, the scope of patch $r_2$ is different from patch $r_1$. After the feature matching, we treat the $r_2$ region containing the matched points as the adjusted corresponding patch $r_2'$, as shown in Fig. 2.

**Locally spatial constraint.** Two main challenges of object verification are the location disparity of the key points due to the view change of two images and the matching outliers. Fortunately, our saccade function $S$ has separated an object into patches. It offers three benefits. Firstly, because the patch size is smaller than the whole image, a patch is more likely to depict the 2D plane of a 3D object. Locally using the spatial constraint like Eq. (1) can be more robust than the origin SP. Secondly, we can assume the center points of patches $r_1$ and $r_2'$ are correct if they share the same pattern. That is we assume $\mathbf{c}_1 = \mathbf{c}_2 = \mathbf{c}$, where $\mathbf{c}_1$ and $\mathbf{c}_2$ are the center locations of $r_1$ and $r_2'$. Treating $\mathbf{c}$ as the reference point, the disparity of two correct matching key points $\mathbf{p}_2^i$ and $\mathbf{p}_1^i$ is $(M - I)(\mathbf{p}_1^i - \mathbf{c})$, where $I$ is the identity matrix. So Eq. (1) can be rewritten as:

$$F = \sum_{p_1^i \in r_1} [\![ \|(\mathbf{p}_2^i - \mathbf{p}_1^i) - (M - I)(\mathbf{p}_1^i - \mathbf{c})\| < \epsilon ]\!]. \tag{3}$$

Like [26], we do not consider the in-plane rotations and overturn. We try to infer a threshold of disparity $(\mathbf{p}_2^i - \mathbf{p}_1^i)$ based on $(\mathbf{p}_1^i - \mathbf{c})$ without calculating a correct $M$. Note that the upper bound of $|\mathbf{p}_1^i - \mathbf{c}|$ is half of the patch diagonal length.

Thirdly, because fovea function only focuses on the regional patterns, we can assume there is no unrelated object or occlusion in $r_1$ and $r_2'$. Thus, we use the inlier ratio to replace the inlier number of SP:

$$F = \frac{1}{|r_1|} \sum_{p_1^i \in r_1} [\![ \|(\mathbf{p}_2^i - \mathbf{p}_1^i) - (M - I)(\mathbf{p}_1^i - \mathbf{c})\| < \epsilon ]\!], \tag{4}$$

where $|r_1|$ is the number of local features in $r_1$. This change is to use probability to deal with the uncertainty of feature matching. Although it is difficult to guarantee the accuracy of a specific local feature matching, we expect the clear different inlier ratio distributions between the correct patches and in-correct patches with enough matching samples.

**Implementation.** We simplify Eq. (4) for the easy of computing. For $p_1^i \in r_1$, the expectation location of its corresponding point $p_2^i \in r_2'$ is $\mathbf{p}_1^i + (M - I)(\mathbf{p}_1^i - \mathbf{c})$. Because photos are taken from a restricted range of canonical views, transformation $M$ varies in a restricted space. We try to predict the location $\mathbf{p}_2^i$ based on $(\mathbf{p}_1^i - \mathbf{c})$ without calculating a correct $M$. To do so, we divide $r_1$ and $r_2'$ into 3 by 3 sub-patches and each sub-patch contains some points. That is $r_1 = [a_1, a_2, ..., a_9]$ and $r_2' = [b_1, b_2, ..., b_9]$, where $a_k$ and $b_k$ are correspondence sub-patches. For a matching point pair $p_1^i \in r_1$ and $p_2^i \in r_2'$, if $p_1^i$ locates in $a_k$, we expect its corresponding point $p_2^i$ will be in $b_k$. In this way, we simplify the Eq. (4) as:

$$F(r_1, r_2') = \sum_{k=1}^{9} f(a_k, b_k), f(a_k, b_k) = \frac{1}{|a_k|} \sum_{p_n \in a_k} k(p_n, p_n'), \tag{5}$$

where $p_n$ and $p_n'$ are two matched points, $k(p_n, p_n') = 1$ if $p_n' \in b_k$, otherwise $k(p_n, p_n') = 0$.

## 4 Experiment

Table 1: Results (% mAP) on the ROxf/RPar datasets and their large-scale versions ROxf+1M/RPar+1M, with both Medium and Hard evaluation protocols.

| Method | ROxf | | ROxf+R1M | | RPar | | RPar+R1M | |
|---|---|---|---|---|---|---|---|---|
| | M | H | M | H | M | H | M | H |
| compare inference ability using SIFT | | | | | | | | |
| VLAD [19] | 37.6 | 21.7 | 31.4 | 10.7 | 83.2 | 69.1 | 69.5 | 44.7 |
| ASMK [42, 48, 45] | 49.6 | 29.1 | 41.4 | 18.0 | 83.6 | 69.5 | 66.2 | 44.4 |
| SP (with ratio test) [28, 26] | 64 | 42 | 56.3 | 30.5 | 86.3 | 71.6 | 70.1 | 46.4 |
| SP (without ratio test) [28, 26] | 44.4 | 23.3 | 37.3 | 13.5 | 83.7 | 69.0 | 66.9 | 44.2 |
| GCRANSAC [5] | 63.3 | 40.2 | 55.7 | 29.7 | 85.9 | 71.0 | 69.6 | 45.7 |
| TP with SP (Ours) | **68.6** | **46.3** | **59.2** | **33.9** | **86.6** | **71.7** | **70.5** | **46.6** |
| TP without SP (Ours) | **69.5** | **46.9** | **60.4** | **34.4** | **86.5** | **71.6** | **70.6** | **46.8** |
| vs. pre-trained models | | | | | | | | |
| CLIP (RN101) [30] | 46.4 | 21.9 | 39.4 | 12.5 | 85.1 | 70.0 | 68.9 | 45.7 |
| CLIP (ViT-B/32) [30] | 47.5 | 24.0 | 38.0 | 13.2 | 83.9 | 68.9 | 69.2 | 46.1 |
| Superpoint [10] | 66.1 | 42.4 | 59.4 | 33.4 | 85.9 | 71.0 | 70.0 | 46.1 |
| vs. large-scale fine-tuned models | | | | | | | | |
| GeM [29]+CL [29] (GLD) [46] | 71.0 | 49.1 | 57.1 | 30.3 | 86.6 | 72.4 | 70.2 | 46.6 |
| GeM [29]+AP [32] (GLD) [46] | 69.9 | 48.2 | 55.1 | 29.1 | 86.8 | 72.7 | 70.5 | 47.1 |

### 4.1 Experiment setting

In our study, we contrast our novel ToPological verification (TP) approach with established methods like SP, Vector of Locally Aggregated Descriptors (VLAD) [19], and Aggregated Selective Match Kernel (ASMK) [42], employing standard SIFT features for the comparison. For SP, we utilize both standard RANSAC and Graph-Cut RANSAC (GCRANSAC) [5], recognized as one of the superior RANSAC methods. The conventional comparison for SP involves reranking the top 100 images found by ASMK using the same local feature [28]. As advanced initial ranking methods can introduce more challenging images and elevate the difficulty [28], we opt for the state-of-the-art fine-tuned feature DELG [7] over SIFT for initial ranking. For fairness, all methods rerank the top 100. Though our primary focus is not on enhancing representative learning, we contrast our method with state-of-the-art (SOTA) pre-training and fine-tuning techniques to underscore the importance of this inference direction and its potential for future exploration. We benchmark our method against pre-trained methods CLIP [30] and Superpoint [10]. In terms of the fine-tuning approach, we choose methods fine-tuned on the extensive, cleaned Google Landmark Dataset (GLD) [46]. We notice that some recent studies claim that GLD is not a fair fine-tuning set for ROxford and RParis because there

are overlapped landmarks [45]. We agree with this claim. However, we still compare our approach with SOTA methods fine-tuned on GLD because our method does not need any fine-tuning, and the methods fine-tuned on GLD, although have fairness issues, show the best performance on ROxford and RPairs. We compare our method with the well-known GeM [29], fine-tuned on GLD using contrastive loss (CL) and AP loss (AP) [32]. To demonstrate the generalizability of our TP, we pair it with the SOTA fine-tuned features of DELG [7] and the SOTA re-ranking method Hypergraph Propagation (HP) [3]. We notice that there are existing claimed better results than DELG and HP on ROxford and RParis, but combining DELG and HP is the best baseline we can successfully reproduce by the submission date of this paper.

## 4.2 Quantitative result

As depicted in Table 1, our method notably surpasses SP, GCRANSAC [5], ASMK, and VLAD across all settings. It significantly outperforms SP by 8.6% and 11.7% on the challenging ROxford under middle and hard settings, respectively. This underlines the feasibility and effectiveness of our novel adaptation of the topological model in RANSAC. Given the widespread usage of SP [26, 33, 35, 34] and its competitiveness with other instance recognition approaches in terms of accuracy [28, 7, 24], this improvement is particularly noteworthy. Our results represent the top non-fine-tuning performance on ROxford and RParis. While GCRANSAC excels at predicting homography or fundamental matrices [5], it does not surpass standard RANSAC in our recognition task. This may be due to the spatial model's incompatibility with recognition tasks, suggesting that enhancing spatial-based RANSAC may not improve landmark

Table 2: Results (% mAP) of HP on the ROxford.

| Method | M | H |
|---|---|---|
| SP (hop1) | 80 | 61 |
| Our TP (hop1) | **81** | **63** |
| SP (hop2) | 85 | 69 |
| Our TP (hop2) | **86** | **71** |

recognition performance. In an ablation study conducted to assess the impact of using SP to filter the hypothesis set, we found that incorporating SP surprisingly led to a decrease in final accuracy. Despite our Python-based method averaging 1.23s per image pair, slower than C-implemented SP at 0.53s, we anticipate considerable speed improvements with a C-based implementation of our method.

Our TP outperforms notable pre-trained models, including CLIP [30] and Superpoint [10], without requiring any training. Although CLIP delivers fair performance, it falls short of both SP and TP. These results underscore the competitive edge of the hypothesis-testing strategy for landmark recognition in highly noisy situations, even against large-scale pre-training methods. Despite Superpoint's superiority to SIFT in SP and its enhancement of inference performance, it remains less effective than our TP. These results attest to our TP's ability to circumvent the inherent issues of the spatial model for recognition.

Our novel TP method demonstrates competitive results against the renowned GeM [29], fine-tuned on the large cleaned Google landmarked dataset (GLD)[46], without any pre-training or fine-tuning. This comparison, while unconventional due to TP being a reranking method and GeM an initial ranking method, is meaningful. GeM, a state-of-the-art retrieval baseline, marked a milestone as the first metric learning method to surpass non-fine-tuned SP approaches [28], following a series of improvements from selective search[41] to R-MAC [43]. Since then, non-fine-tuning techniques have seen limited advancements. However, TP breaks this trend, achieving

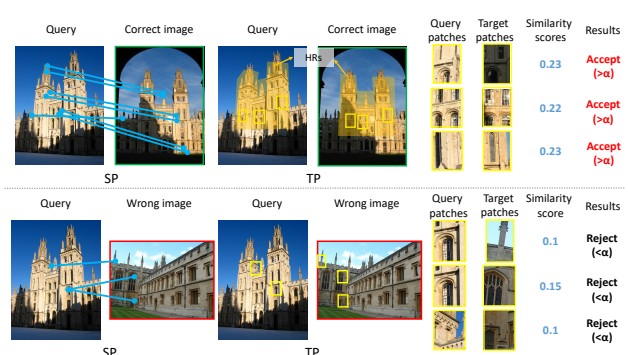

Figure 3: SP results, TP results, and some TP steps for verifying a correct image pair (upper) and a wrong image pair (down) using SIFT features. The threshold $\alpha$ is set as 0.2.

comparable results to GLD-trained GeM. This is not to say SIFT is enough for this task but

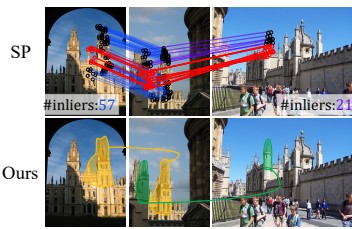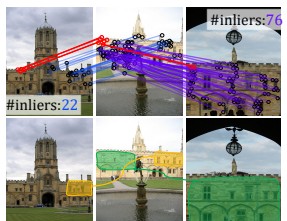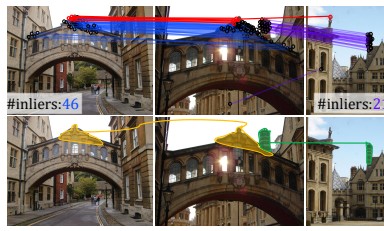

Figure 4: Hypergraph propagation results using traditional SP (upper) and our method (down). In each triplet, the first image and third image are different. Red lines show SP's wrongly connected local features between the first image and the third image. The orange and green regions show the verified HRs between each image pair. Our method correctly separates the local features of the first and the third images. HR is the union of many verified patches. We only show the outline of key points in HR for ease of observation.

to emphasize the potential of hypothesis testing when combined with a topological model for non-fine-tuning image retrieval.

We acknowledge that advancements in fine-tuning features for ASMK or SP, such as SuperFeature and DELG [45, 7], have emerged recently. While it is possible to train features for our TP, we leave this for future work, as this work mainly focuses on the non-fine-tuning retrieval. However, to show our method's generalization capabilities, we still apply TP to fine-tuned DELG local features and Hypergraph Propagation (HP) [3]. HP is a fast reranking method that combines verification and diffusion on local features. It runs the heavy verification offline and thus achieves fast, low-memory, and high-accuracy reranking online. Table 2 demonstrates that our method outperforms SP on HP, irrespective of propagating among hop1 or hop2 neighbors; hop1 or hop2 is the truncation hyperparameter in diffusion [3]. This improvement is noteworthy, considering it does not necessitate any additional training. This is the SOTA retrieval performance on ROxford to the best of our knowledge.

We understand that readers may have concerns about the **practical value of a RANSAC-based method** in retrieval, given its slower processing time compared to using dot products with learned embedding vectors. However, it is important to note that if RANSAC can provide reliable labels without fine-tuning, these labels can be used directly to train the embedding space for any database, eliminating the need for a specially collected fine-tuning set. As we shown in Table 2, it can also directly improve HP accuracy. Because verification in HP is offline, the slower speed of RANSAC is acceptable. Furthermore, SP is employed not only in retrieval but also in other tasks on mobile devices where model size is crucial, such as loop-closure detection. Our TP, which shares the lightweight nature of SP without involving millions of parameters like models such as ResNet [15], offers enhanced robustness and accuracy. Most importantly, our method is highly explainable, which we will discuss in the following section.

### 4.3 Qualitative result

In addition to its accuracy, our method exhibits high explainability, a crucial aspect for real-life applications. The explainability of our method can be understood in three aspects:

1. We can observe how the HRs are constructed step by step, find a correspondence patch for each patch in HR, and check how each candidate patch pair is accepted or rejected by the fovea function $F$, as shown in Figs. 2 and 3. These characters offer a clear understanding of how our method works.

2. The TP detected HRs and matched patches are easy to comprehend, whether using SIFT (Fig. 3) or DELG (Figs. 1 and 4). As shown in the upper row of Fig. 3, although SP can also give a high inlier number for the correct image pair, its found matchings can be wrong. Our TP gives more reasonable results. This issue of high inlier counts with incorrect matchings is also observed in DELG, suggesting that fine-tuning can improve the similarity of local features for the same object, but cannot guarantee the accuracy of their locations. Additional examples are provided in the supplementary material. Fig. 4 compares the qualitative performance of SP and our TP on HP. The first query and third image in each triplet do not depict the same building. Due to SP's erroneous matching of some local features (indicated by red lines), a search engine using SP mistakenly identifies the third

image as a positive match. In contrast, our method accurately matches the correct similar regions between two images and can correctly determine that the third image is unrelated to the query, thereby enhancing retrieval performance.

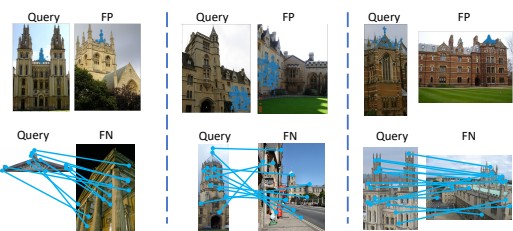

Figure 5: The false positive (FP) and false negative (FN) cases of our method. We draw the SIFT feature locations in the largest detected HRs for FP cases and the ratio test results for FN cases. Check more examples in the supplementary material. Check all the FP and FN cases from this link.

3. Our approach maintains explainability, even in cases of false positives and negatives, offering valuable insights into the effectiveness and potential issues at each stage. We have visualized all false negatives (FN) and positives (FP) in Fig. 5. For a more comprehensive review, we provide a link for readers to explore further. Interestingly, without context, many FP cases are challenging for the authors to distinguish, signaling the effectiveness of our fovea function F in generating reasonable patch similarity results. Upon examining FN cases, we recognize areas for enhancement. The primary issues lie in incorrect ratio test results and significant size changes. While our method considers size changes, it struggles with handling extreme variations. An easy solution is to multi-rescale each image and traverse all possible matching scales.

We refrained from this to avoid significantly reducing speed and to maintain a fair comparison with SP. However, future work could potentially apply this strategy or devise a better sampling method to manage extreme scale changes, thereby improving performance.

## 5 Discussion about the task value

This paper aims to advance explainable image retrieval in scenarios where a fine-tuning set is unavailable. This pursuit carries significant weight as acquiring a large fine-tuning set can often be impractical or costly, especially in open-world or private settings. Therefore, research in this domain has the potential to enhance the utility of AI systems across a variety of real-world contexts.

Our exploration into non-fine-tuned retrieval is further motivated by the observed gap between human-level performance and that of the state-of-the-art (SOTA) fine-tuned models [4, 40] on challenging datasets such as ROxford/ RParis [28] and PROxford/ PRParis [3]. Notably, these datasets provide invaluable user study results, as each positive image pair is verified to be identifiable by humans without the need for contextual visual information [28, 4]. Our recent study shows that annotators who are not familiar with European buildings can recognize, localize, and segment the target landmark of all the positive image pairs in ROxford and RParis. We propose the pixel-retrieval task and its first benchmarks PROxford and PRParis, to evaluate the pixel-level recognition [4]. Interestingly, humans achieve this remarkable recognition ability without extensive in-domain training, such as learning from the Google Landmarks Dataset (GLD) [46]. Yet, even the best fine-tuned models [7, 40] trained on the vast GLD with 4.1 million images have yet to reach human-level accuracy on ROxford/RParis and PROxford/PRParis. This discrepancy prompts us to question how humans excel in such recognition tasks without large-scale fine-tuning.

The process of recognition is a confluence of perception and cognition. The superior instance recognition of humans in the ROxford dataset might be more cognitively inclined, though concrete evidence remains elusive. Cognition entails reasoning based on existing knowledge for decision-making and the discovery of new knowledge [47, 13, 37]. When summarizing notable human instance recognition experiments [6, 16, 39], Greyson [1] defines perception as the direct link between an instance and its visual features, while cognition involves understanding the object's changes in viewpoint or state (*e.g.*, background and illumination) [47, 1, 11, 22]. Machine learning methods that align with this definition of cognition include ASMK [42] and SP [26], which further motivates us to explore improvements to SP without fine-tuning. Future research discerning the nuances between perception and cognition could potentially enhance non-fine-tuning methods for real-world applications.

# 6 Conclusion

Our study highlights the limitations of the SPatial verification (SP) method, particularly its reliance on a spatial model for recognition. As a potential alternative, we propose the use of topological common sense, which significantly outperforms SP and achieves unprecedented performance in non-fine-tuning instance recognition. This underscores the potential of hypothesis-testing inference.

The inspiration for our saccade and fovea function comes from the human process of comparing two newly seen images. In this sense, our topological approach aligns more closely with the human visual system than the spatial approach. While humans don't compute transformation matrices like SP when comparing images, they intuitively understand the topological relations among observed objects—for instance, recognizing that a hand is connected to an arm. Notably, topological information is vital to the human visual system, as it is consistently preserved even when visual signals undergo significant deformation during transmission from the retina to the lateral geniculate nucleus and visual cortex [44, 18].

Importantly, our method doesn't compete with existing fine-tuning methods but complements them. It can directly integrate learned features, and future work could even fine-tune features specifically for this topological approach.

While our results are promising, they also pave the way for future exploration. Subsequent work could investigate the incorporation of more complex common sense concepts, both explicit, like our topological model, and implicit, derived from pre-training. Enhancements in these areas could potentially boost the robustness and adaptability of our method. Further study into the convergence properties of our method and refinement of the hypothesis sampling strategy are also warranted. Through addressing these areas, we aspire to advance object recognition and highlight the potential of hypothesis-testing approaches in artificial intelligence.

## Acknowledgments

We would like to thank the anonymous reviewers for their valuable comments and insightful suggestions. Sung-Eui Yoon is the corresponding author of this paper. This work was supported by the Institute of Information & communications Technology Planning & Evaluation(IITP) grant funded by the Korea government(MSIT) (RS-2023-00237965, Recognition, Action and Interaction Algorithms for Open-world Robot Service) and the National Research Foundation of Korea(NRF) grant funded by the Korea government(MSIT) (No. RS-2023-00208506).

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
