# A    Additional qualitative results

To enhance our understanding of the ToPological verification (TP) performance, we provide a detailed visualization analysis. We begin by illustrating successful verification results in Appendix A.1, showcasing the efficacy of our approach.

Next, in Appendix A.2, we present standard verification results. While our method may not perfectly align with every correspondence patch, it accurately identifies and produces substantial Homeomorphism Regions (HRs). These HRs are integral in ensuring the correct index images are ranked highly in the final retrieval list. To further contextualize our TP's advantages, we juxtapose these standard verification results with those of the SPatial verification (SP). This comparison underscores how our method outperforms SP, delivering superior results.

Lastly, we evaluate the limitations of our method by visualizing all false positive (FP) and false negative (FN) results in Appendix A.3. These examples not only highlight the extreme cases encountered by our method but also illuminate potential future research directions.

Through this comprehensive visualization analysis, we demonstrate the robust performance of our TP method and its clear superiority over the SP method.

## A.1    Good verification results

Figure A3's upper segment showcases the impressive outcomes of our topological verification. We illustrate the identified Homeomorphism Regions (HRs) within the correctly matched image pairs. HRs encompass a multitude of verified patches; for visual clarity, we've outlined the SIFT points in the HRs. In every pair, the identified HRs effectively cover a significant proportion of the query object region. Consequently, our method accurately assigns high similarity scores to these image pairs, positioning the index images at the forefront of the retrieval results.

## A.2    Standard verification results: compared with SP

The lower portion of Figure A1 exemplifies instances where our method identifies the correct, relatively substantial HRs for each accurate image pair. Though these identified HRs might not match the superiority of the results in Appendix A.1, they correctly envelop each image pair's corresponding regions and are comparatively larger than the HRs detected in the false positive pairs illustrated in Figure A2. Hence, our method suitably ranks these accurate index images highly.

We further evaluate our topological verification outcomes against those of the SP method. As shown in the lower part of Figure A1, the SP method often mismatches points within each image pair, leading to erroneous results. Conversely, our TP method accurately identifies and matches the correct correspondence regions, demonstrating its superior performance.

## A.3    False positives and false negatives

In addition to successful verification instances, we also explore cases where our method fails. We present visualizations of **all** the false positive (FP) and **all** the false negative (FN) Homeomorphism Regions (HRs) identified by our method on ROxford. Figures A2 and A3 offer some examples, and all the results can be accessed via this link.

In false positive instances, our method detects multiple HRs for each pair, with the SIFT feature locations drawn in the largest detected HR. Despite the distinct objects within each image pair, we note remarkable similarities within the detected HRs. In fact, focusing solely on these HR regions, without considering other image aspects such as object shape, even we find difficulty distinguishing many of these regions.

Regarding false negative cases, our method fails to detect any HRs. Figure 5 in the main body of this paper and this link display the ratio test matching results. We recognize limitations in our method's ability to handle extreme size changes. This prompted us to investigate whether the SP might yield better results for these cases.

We subsequently illustrate some SP matching results in Figure A3. It becomes evident that SP also struggles with these challenging scenarios. While neither TP nor SP can effectively tackle

Detected HRs

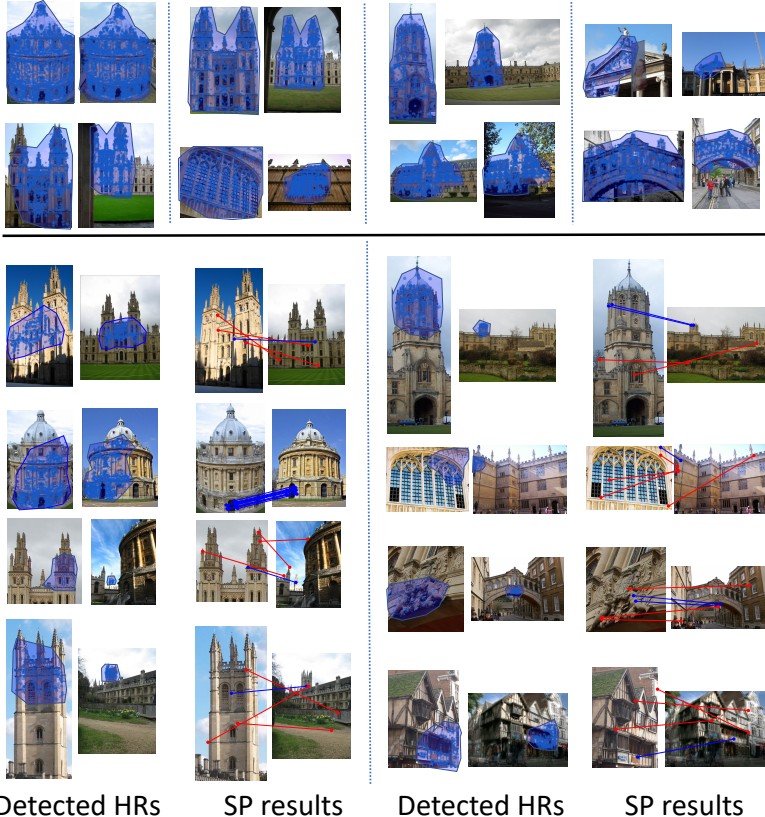

Detected HRs    SP results    Detected HRs    SP results

Figure A1: The upper section presents successful verification outcomes achieved by our method, while the lower section displays standard verification results, contrasted against SP's performance. Each image pair consists of a query image on the left and an index image on the right. Blue points signify SIFT features found within the detected Homeomorphism Regions (HRs), and blue polygons outline the borders of these HRs. Connections depicting correct correspondences as a result of the SP method are shown with blue lines. Conversely, red lines represent SP connections that link incorrect locations.

these problematic cases, they are not entirely intractable. As mentioned in the main body of this paper, implementing multi-rescaling for each image and exploring all potential matching scales could provide a solution. However, such an approach would significantly increase computational costs. A more efficient alternative could involve the development of an advanced sampling method to better handle extreme scale changes.

## B  Explanation to fovea and saccade functions

In Section 4.3 of the main document, we delve into the explicability of our Homeomorphism Region (HR) construction procedure as part of our ToPological verification (TP) process. Each patch in the HR can be mapped back to its analogous patch, providing insight into how the fovea function $F$ either accepts or rejects every candidate patch pair. Some examples of this process were illustrated in Figure 3 of the main text. Further demonstrations of this method can be found in Figures A4 and A5 in this supplement.

Given a pair of images and a hypothesized translation, our method iteratively generates and verifies potential patch pairs. Key intermediary steps for both correct and incorrect patch pairings are visualized in Figures A4 and A5. Upon reviewing these steps, we observed that the function $F$

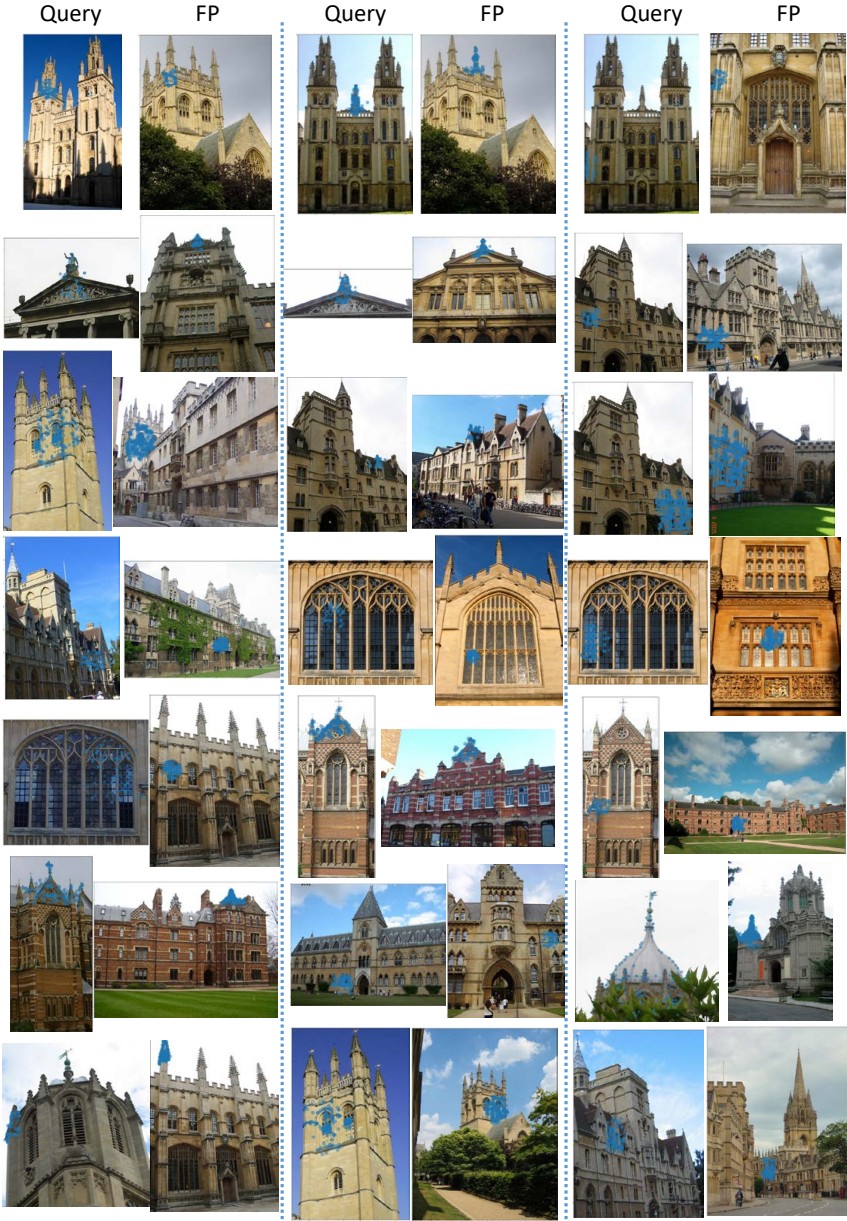

Figure A2: False positive (FP) image pairs. We draw SIFT features inside the largest HRs for these false positive image pairs. Although the images in each pair have different buildings, the detected HRs are similar with each other. Check all of them from this link

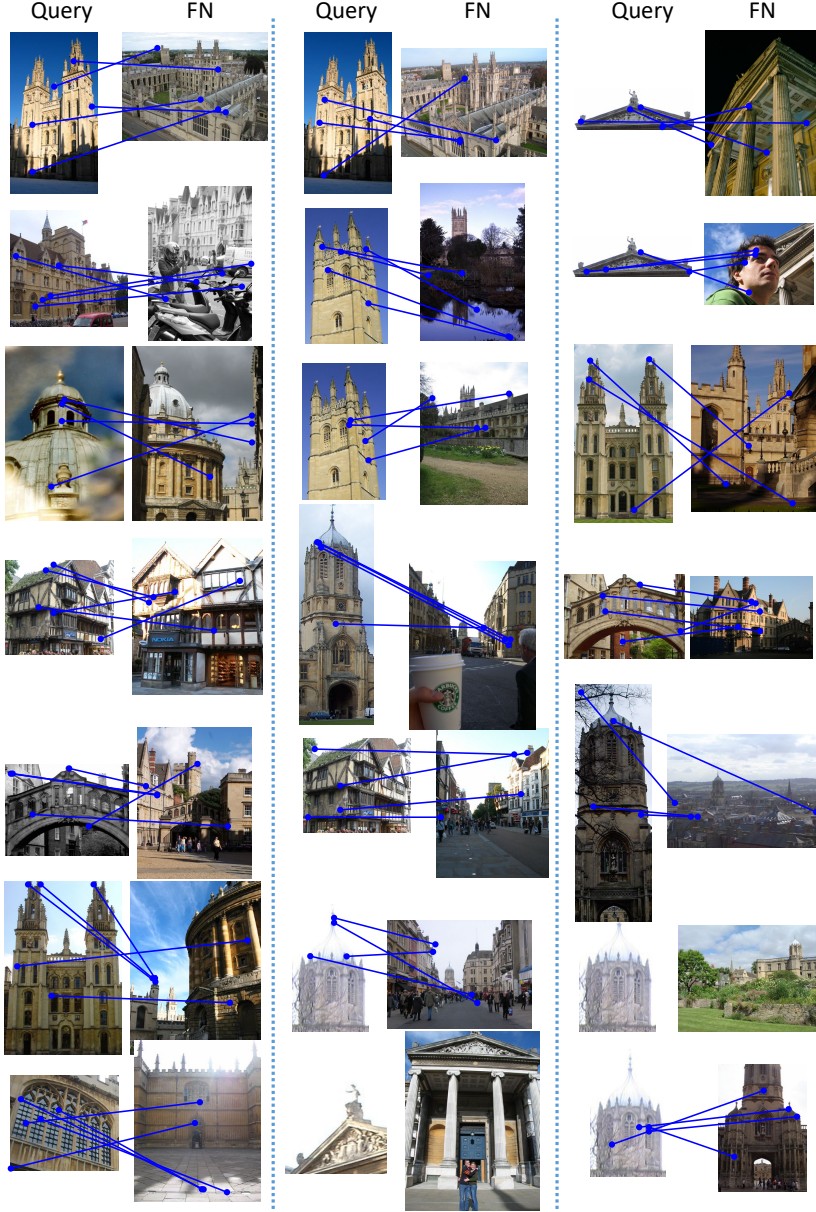

Figure A3: Image pairs categorized as false negatives (FNs) where our method fails to detect any Homeomorphism Regions (HRs). In addition to this, Figure 5 in the main body of the paper, along with the provided link, offer visualizations of the ratio test results for these pairs. We supplement these illustrations with SPatial verification (SP) results, revealing that the SP approach also fails to yield accurate results in these instances.

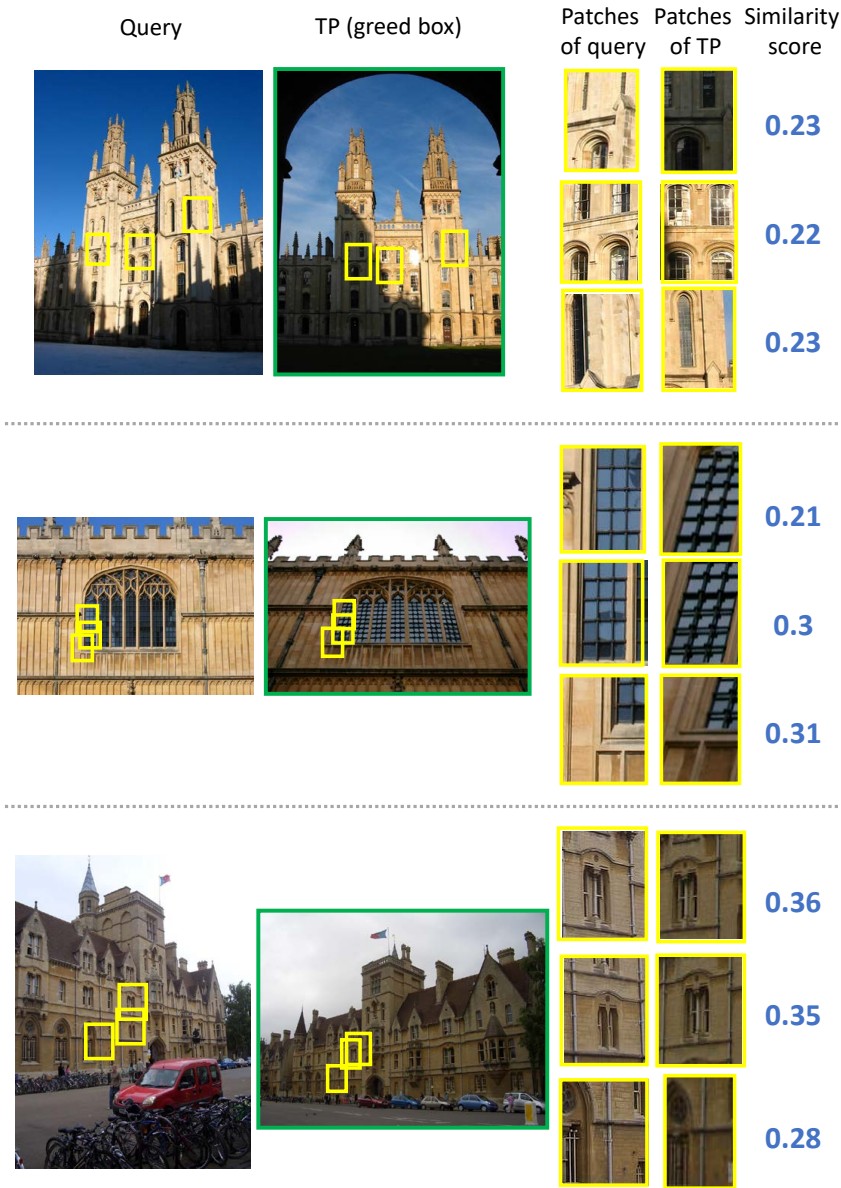

Figure A4: This figure presents the similarity scores for the correctly matched patch pairs throughout the verification process. These pairs are suggested by the function $S$ and subsequently authenticated by the function $F$. Our method appropriately assigns high scores to these similar patches, with their similarity scores surpassing the threshold of 0.2.

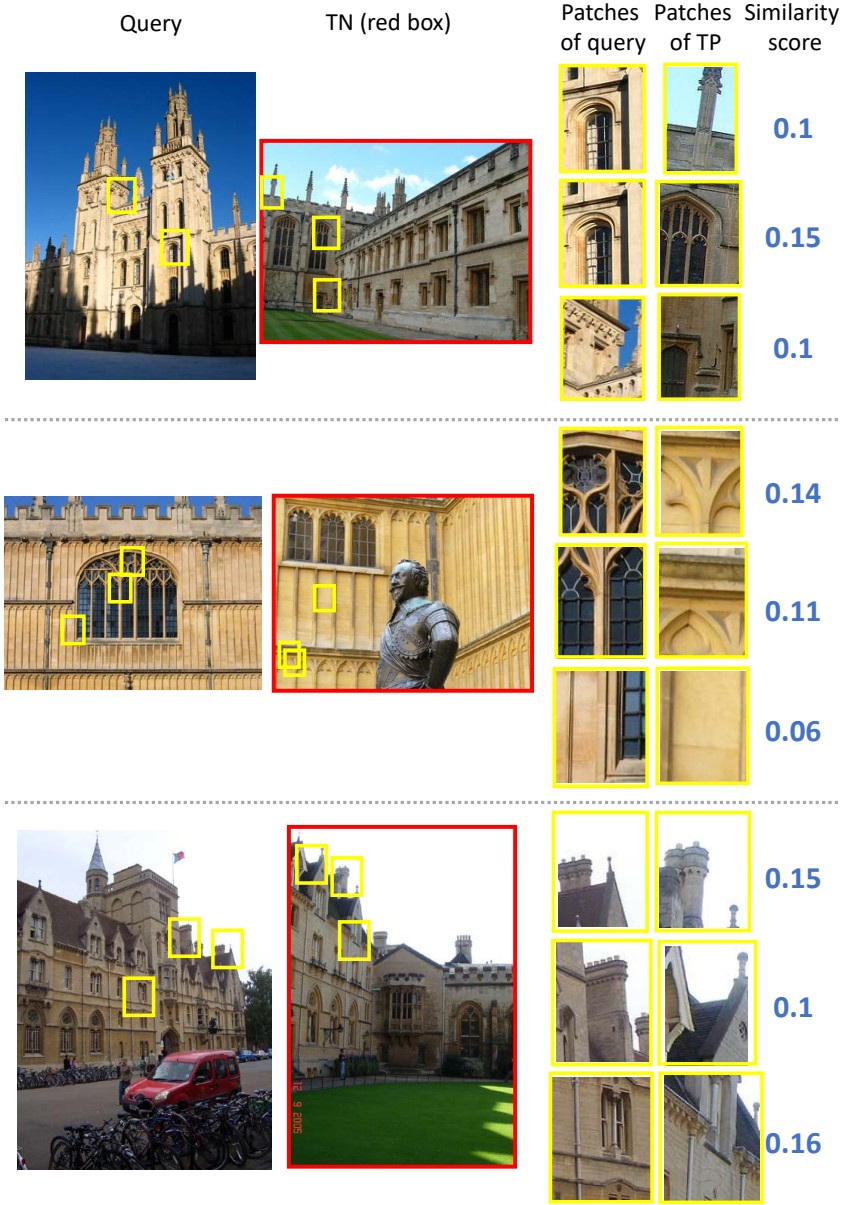

Figure A5: This figure displays the similarity scores for the incorrectly matched patch pairs during the verification process. These improper pairings are derived from ratio-test matching and serve as our hypotheses. Our method successfully allocates low scores to these dissimilar patches, with their similarity scores falling below the threshold of 0.2.

accurately computes patch similarity. By implementing a threshold of 0.2, the function effectively differentiates between correct and incorrect patches as demonstrated in Figures A4 and A5.

## C  Combined with fine-tuned features

In our research, we've pinpointed two principal challenges intrinsic to SPatial verification (SP): the presupposition of planar structures and the oversight of topological relationships among features. Both of these challenges can be also be alleviated through fine-tuning using the same domain data.

The first challenge refers to the assumption that features exist on planar structures. We can alleviate this by elevating the distance disparity, symbolized as $\epsilon$ in Eq. (2) of our primary document, and by enhancing the alignment of correspondence features. The second concern involves the omission of topological relationships among features. This issue can be mitigated due to the extensive receptive field of each feature based on Convolutional Neural Networks (CNN). This breadth of the receptive field empowers CNN to autonomously learn the topological relationships, bridging the connection between the central pattern and the other regional patterns.

However, this method heavily relies on fine-tuning and may not prove effective when compared to our proposed ToPological verification (TP) in scenarios where no same domain fine-tuning set is accessible.

This paper does not primarily focus on refining fine-tuning features, but we nevertheless tested the generalizability of our TP on the fine-tuned features of DELG. As stated in the main body of the paper, we discovered that our TP could enhance the retrieval results of DELG. While the fine-tuned DELG often provides a high inlier count for correct image pairs, it doesn't necessarily guarantee the precision of the matching locations. As illustrated in Figure A6, the matching locations of SP are not always precise, sometimes resulting in incorrect location matches. In such instances, our TP connects only the correct patches, demonstrating its practicality and effectiveness.

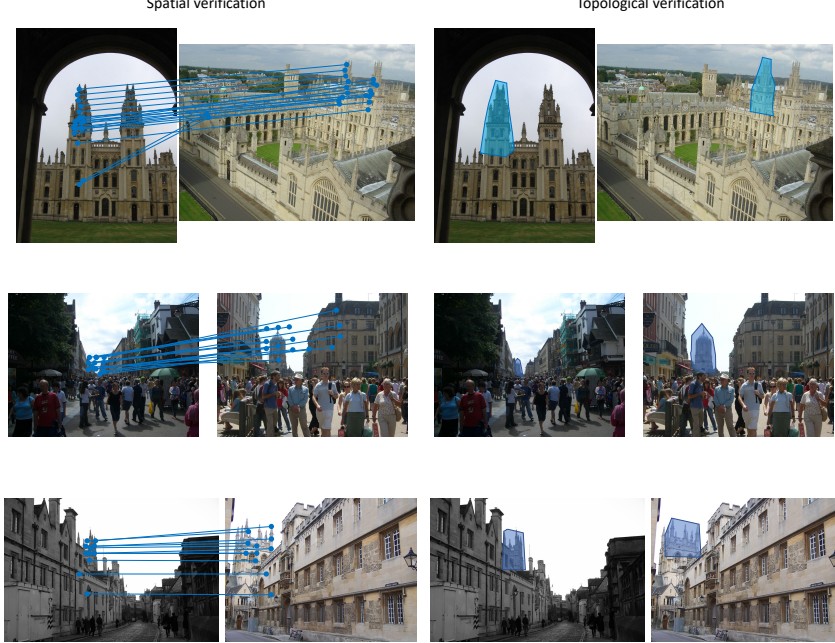

Figure A6: This figure presents a comparison between spatial verification and topological verification, utilizing the fine-tuned DELG. In some instances, spatial verification produces matchings that lack precision. In contrast, our topological verification method exclusively forms connections between correct patches, illustrating its accuracy and effectiveness.