# OpenReview forum: "Topological RANSAC for instance verification and retrieval without fine-tuning"
_NeurIPS.cc/2023/Conference — NeurIPS 2023 poster_

### Official Review · Reviewer_JDZk · 2023-07-06

**Soundness:** 3 good
**Presentation:** 2 fair
**Contribution:** 3 good
**Rating:** 5
**Confidence:** 4

**Summary:**

This work introduced a new approach of geometrical verification for image retrieval and matching, which is not based on pixel-perfect robust estimation, but on something called "topological common sense".  The authors discuss the limitations of the commonly used Spatial Verification strategies, and draw inspiration from how humans compare two images (topological relations vs point level accurate estimation of transformation matrices). Good improvement is shown in ROxf/RPar datasets.

**Strengths:**

- TP is a very interesting idea compare to SP.
- The biological inspiration for TP is good, and sound.
- The fact that "TP without SP (Ours)" outperforms "TP with SP (Ours)", is a good indication of the robustness of the proposed method.
- The high explainability experiments are interesting and seem technically accurate.

**Weaknesses:**

- Why is it claimed that this method offers explainability? To me it seems like the SP methods also offer explainability. Can the authors explain?
- Why was retrieval only chosen as a proxy task? It seems like TP could be used in other tasks too (e.g. matching, loop closure) which would make this work much more general and thus much stronger.
- It seems like Hypergraph propagation results are not ideal when using SP. I am wondering if there is any extra loss or consistency regularizer that could be added to SP to make it avoid this. In a sense, is this a fair comparison? Are all pairwise pose estimation consistency losses get into account for the 3 image case? {e.g. L63 RANSAC, SP calculates a transformation matrix M between two views, but is that enforced somehow in the triplet?}

**Questions:**

My questions are above in the Weaknesses section, and these are the things I would like to see answered from the authors.

**Limitations:**

I did not find that the authors addressed limitations and potential negative societal impact of their work.

---

> ### Author Rebuttal · Authors · 2023-08-09
>
> Dear Reviewer JDZk,
>
> Thank you for acknowledging the significance of our method and recognizing its merit in terms of its robustness and technical accuracy. We wish to address the concerns you've raised:
>
> Explainability: We'd like to emphasize that we aren't suggesting our method, TP, offers superior explainability in comparison to SP, which is well-regarded for its strong explainability. Instead, our contention is that TP offers improved explainability than other methods that augment SP and often rely heavily on fine-tuning sets and compromising on clarity and transparency. If we were to identify a specific advantage our TP possesses over SP in terms of explainability, it would be its user-friendly reasoning. The HRs generated by TP align more intuitively with human understanding, contrasting with SP's inlier numbers. Without the context provided by the homography matrix, these numbers can sometimes puzzle users in discerning why certain matchings are selected or discarded.
>
> Task Specificity: Our TP is designed to target the rigid-body recognition challenges. The rationale for leveraging retrieval as the proxy task stems from the notorious complexity of datasets like Oxford and Paris. These datasets encompass hard positives and negatives, which while being challenging, have ground truths verifiable by human annotators. As concurred by Reviewer WbBp and discussed in Section 5, the ability of humans to discern these tough cases underscores the importance of our target task. Conversely, other rigid-body benchmarks don't match the complexity of Oxford and Paris or guarantee human recognition. For instance, the geo-localization benchmark MSLS uses images within a 25m radius as ground truth indicators. Regrettably, some pairs lack visual overlap, making human recognition based solely on visual cues unfeasible.
> Further emphasizing on this, we evaluated our approach against 100 queries from the challenging geo-localization benchmark MSLS val set. As depicted in Table 2 in the rebuttal PDF, our method shows marked improvement over SP with SIFT features. Surprisingly, our TP even rivals the performance of the recent large-scale fine-tuned model R2Former (CVPR 23); it fine-tuned on more than 1.44 million training images in MSLS. This experiment, we believe, reinforces our method's versatility, as we already showed in the submitted paper in page 7 and table 1 w/ GeM fine-tuned on GLD 4 million training set.
>
>
> Fairness in Comparison with SP on Hypergraph Propagation: Our commitment to fairness is evident from the consistent use of code, weights, and pre-calculated SP results employed by the hypergraph propagation (HP) research. Given that our parameters mirror those perfected by the HP authors (and remain the best performance in ROxford/RParis benchmarks), it is reasonable to think that the current setting for SP on HP is good and the comparison stands equitable. The sole modification was to change hop 3 to hop2 to reduce computational overhead. We guess your concern about fairness is because the listed performance in our paper is lower than the performance of the HP paper. To address any persisting concerns, we conducted additional tests comparing SP and our TP with propagation set at hop 3 in Table 3 of the rebuttal PDF. We reproduced the reported performance in their paper and our TP outperformed their results, further validating our method's efficacy.
>
> We hope these clarifications address your concerns, and thank you once again for your constructive feedback.
>
> Warm regards,
>
> Authors

---

### Official Review · Reviewer_pXos · 2023-07-06

**Soundness:** 3 good
**Presentation:** 2 fair
**Contribution:** 2 fair
**Rating:** 6
**Confidence:** 3

**Summary:**

To address the limitations of SPatial verification (SP), the authors introduce the topological consistency to the RANSAC process for image retrieval without fine-tuning. With the socalled homeomorphism regions, the proposed method can achieve better results than some typical methods on four datasets.

**Strengths:**

1. This work introduces the topological consistency to the RANSAC process. I think it is interesting.
2. The proposed methods are largely based on existing works. Thus, most of the techniques are correct.
3. The performance of the proposed methods is better than some typical methods.

**Weaknesses:**

Some key concerns should be addressed.
1. Technical contributions

In fact, the methods of comning RANSAC and topological models are not new in current CV field, especically in image retrieval. I agree with that the SP method has big problems without finetuning sets. However, the authors mainly combines the so-called topological  model and RANSAC. Most of other modules are not new. The defination of HR is common sense. There are no new insights. In my view, this is not a significant contribution, but only a typical combination method for better matching feature representations. Besides, as far as I know, other methods without finetuing can also perform better than the proposed method, such as BLIP, BEiT, ect. The proposed functions in Fig.2 are also naive for HRs. Why them should be that order? There are no full motivation.Thus, the techinical contributions are rather limited, considering most of the used modules are from typical image processing operation.

2. Insufficient experiments

First, through the whole experiments, I find that there are no full comparisons with other methods. In fact, the authors only compare with limited methods as shown in Tab.1. As for the methods using SIFT, I suggest to add more comparisions in this aspect. For the pre-trained models, more large-scale models should be added such as BLIP, BEiT, etc. Second, since the title is for image retrieval, I think it is better to perform experiments on different kinds of object retrieval datasets, such as person (Market1501, MSMT2017), Cars (VehicleReID)， Food (Food100k), etc. Third, there are no full analysis on the key modules. For example, how about without using HRs, or just using the visually similar regions? The effects of the proceduce or operation orders in Fig, 2 is missing? As far as I know, there are many different mactching techniques. How about applying the proposed methods in GLUE, SuperGLUE? Is the deep features important in the proposed framework? l think, more experiments will make the proposed method more convincing.

**Questions:**

Please see the weakness.

**Limitations:**

I think the authors have partly addressed the limitations. However, the speed and computation should be discussed for better future applications.

---

> ### Author Rebuttal · Authors · 2023-08-09
>
> Dear reviewer pXos,
>
> Thank you for your meticulous feedback. We understand and acknowledge your concerns and have accordingly undertaken additional experiments and provided clearer explanations.
>
> # Main Contribution and Novelty
>
> Our contribution lies in our innovative adaptation of RANSAC. We replace its geometry model with a topological one, rather than a mere fusion. The essence lies in its innate adaptability to any input pair, functioning autonomously, without the necessity of training data.
>
> While notable methods such as SuperGlue and R2Former (CVPR 23) utilize topological insights, they do not intrinsically modify RANSAC's architecture. Our method, unlike them, excels without a foundational reliance on prior training data. Referring to Fig.2, it portrays only a single loop iteration in RANSAC, which is variable and not predefined. Page 4 should have clarified this; however, we can provide further elaboration in the discussion stage if required.
>
> Please note that feedback from other reviewers supports our standpoint:
>
> "Very interesting" - JDZk.
> "Very different than traditional approaches and refreshing" - X5kj.
> "Brings many insights and advantages and revitalized the classical algorithms" - WpBp.
>
> # Scope and Motivation
>
> Our work delves deep into hypothesis-testing processes that function independently of training; this direction is supported by Reviewer WpBp as "remarkable considering the recent trend of pursuing stronger features through fine-tuning. As discussed in Section 5, it is evident that as humans, we can easily determine if the given image pair is a mismatch without too much contextual information." The result that our no-training TP is on par with large-scale refined GeM shows the potential of this direction.
>
> Addressing rigid-body instance retrieval, our findings largely hinge on ROxford and RParis benchmarks. However, based on your insights, we've widened our experimental spectrum to include the geo-localization benchmark MSLS.
>
> The term "image retrieval" in our title aligns with prevailing norms, as many landmark retrieval papers predominantly analyzing Oxford/Paris use such phrasing. Nonetheless, we're open to title revisions to ensure clarity.
>
> # Additional Experiments
>
> Responding to your suggestions, we've taken the following measures (all the result tables are in the rebuttal pdf file):
>
> Table 1 additionally compared our methodology with more techniques such as SIFT-based FisherVector and SMK, and pre-trained alternatives like BLIP and the BEiT series. While BLIP and BEiT's performance might seem unexpectedly subpar, challenges intrinsic to instance retrieval benchmarks like Revisited Oxford/Paris, featuring extreme viewpoints, occlusions, and pattern similarities, could be contributing factors. We've rigorously vetted our BLIP and BEiT implementations, confirming their proficiency in identifying easy cases, but recognizing their limitations in discerning hard positives and negatives.
>
>
> Table 2 assessed our approach's efficacy on the MSLS dataset, an intricate geo-localization benchmark. The results, especially when pitted against the best fine-tuned model R2Former (CVPR23), are telling of our method's prowess. Because we only consider the rigid-body instance retrieval, you mentioned datasets are not in the scope of this paper. (We are open to change the title to avoid misunderstanding) However, the results on the famous ROxford, RParis, and MSLS show that our method has generality on both two representative rigid-body recognition tasks, landmarks retrieval, and geo-localization.
>
> To further highlight the effectiveness of our proposed HRs, we aligned it with regions established by acclaimed methods such as superpoint+superglue, superpoint + sp, and r2d2 in Table 1, utilizing their pre-existing weightings without fine-tuning on the landmarks in Oxford and Paris. Despite these methods exhibiting strength in tasks like pose estimation and SfM, they couldn't parallel our TP's efficiency. Finally, we would like to humbly mention that our initially selected baselines are the strong and representative methods in the landmark retrieval field; initial feedback from peer reviewers on our experiments has been particularly encouraging:
>
> “The ablation study is a good indication of robustness” - Reviewer JDZk.
> “Very strong results, well-constructed experimental section, apt baselines, and thorough comparisons” - Reviewer X5kj.
> “The benchmarking outcomes are truly exceptional” - Reviewer WpBp.
>
> In conclusion, your feedback is invaluable, and we genuinely believe our refined explanations and bolstered experimental data should address any lingering concerns.
>
> Warm regards,
>
> Authors

---

> > ### Comment · Reviewer_pXos · 2023-08-19
> > **Feedback for authors**
> >
> > I have read the rebuttal and the review comments. I think the authors have partly addressed my concerns. I would like to improve my score.

---

> > > ### Author Response · Authors · 2023-08-21
> > >
> > > Thank you for revisiting our submission. We appreciate your feedback and the constructive comments you provided in the initial review. We're pleased to note that our rebuttal partly addressed your concerns and that you would like to improve your score. If there are specific areas still needing clarification, please highlight them. We aim to address any residual issues and appreciate your guidance in this matter.

---

### Official Review · Reviewer_WpBp · 2023-07-08

**Soundness:** 4 excellent
**Presentation:** 4 excellent
**Contribution:** 4 excellent
**Rating:** 8
**Confidence:** 4

**Summary:**

This paper reexamines the classical instance recognition problem in computer vision, a problem that holds great significance in applications such as image retrieval. In recent years, data-driven approaches that relying on fine-tuning pre-trained deep models have drawn much attention. However, these methods not only face the difficulties in obtaining the data for fine-tuning in real-world applications, but also lack explainability. The authors closely examined the vulnerabilities of SPatial verification (SP) method, which remains the most performant method in non-fine-tuning setting, and propose the new Topological verification (TP) model to replace SP in the RANSAC process. Inspired by human vision system, TP-based RANSAC is a region-growing algorithm which seeks and maximizes Homeomorphism region (HR). Starting from some seed patches called Hypothesis Set, the *fovea* and *saccade* function iteratively examines neighboring regions and expands HR by enforcing several important conditions, including *local consistency*, *topological consistency* and *connectivity*. Evaluation on challenging benchmarks, such as ROxf/RPar and ROxf+1M/RPar+1M, demonstrates that the proposed method establishes new SOTA performance across all methods without the need for fine-tuning.

**Strengths:**

After a decade with deep learning techniques dominating the CV/ML/AI world, it is refreshing and delightful to read this paper, which revitalizes the classical algorithms by introducing many insights and advancements. The authors delve into the SP model that has been widely used in the hypothesis testing of RANSAC process and identify its vulnerabilities that were long overlooked by the previous methods. This is remarkable considering the recent trend of pursuing stronger features through fine-tuning. As discussed in Section 5, it is evident that as humans, we can easily determine if the given image pair is a mismatch without too much contextual information. It is intriguing to observe how the topological rules introduced in this paper closely emulate the tactics that humans may leverage to solve this problem. The bio-inspired fovea and saccade functions effectively replicate such brain mechanisms in an algorithmic fashion. Compared to the SP model and fine-tuning methods, the topological model possesses better explainability. Moreover, it retains the characteristic of not necessitating fine-tuning while also being compatible with large-scale pre-trained models. The benchmarking results are exceptional. I particularly like the insightful discussions in Section 5. It is a very good read.

**Weaknesses:**

If I have to point out one weakness of this paper, I would say that there are still some missing details (some are listed in the Question section of this review) in the main paper and supplemental material, though most of them are minor and does not cause difficulty in understanding the method. However, I still think that the paper could be made a little clearer for readers who do not particularly work on instance recognition and image retrieval problem. For example, I guess not every reader is familiar with the term *hop1* and *hop2* shown in Table 2.

**Questions:**

1. It is not very clear to me how to obtain the initial hypothesis set? It perhaps has been standardized by previous RANSAC-based SP methods. I still think it appropriate to mention it in the paper for clarity.
2. How does saccade function work? I assume that it is similar to most advancing front techniques, as illustrated by part of Figure 2. I don't seem to find more details in the supplemental material as well.
3. How to perform patch location adjustment? I assume that it is accomplished by computing the minimal enclosing bounding box of the matched key points.

**Limitations:**

Nowadays, the mainstream research seems to lean towards seeking for larger and stronger per-trained models to vectorize everything and simplify the retrieval problem as similarity search by simple operations like dot products. Focusing on right instance recognition, the topological model excels in single modality domain and has little applicability in cross-modality retrieval tasks as popularized by CLIP. On the other hand, inspired by human vision system, this work could also be a great example showing that the pursue of large models might be a detour when solving a cognitive problem.

---

> ### Author Rebuttal · Authors · 2023-08-09
>
> Dear Reviewer WpBp,
>
> Thank you for your positive remarks regarding our work. We truly appreciate your support for the direction of classical RANSAC, especially at a time when it isn't the prevailing trend. We concur with your observation that our findings underscore the potential of RANSAC-based methods when combined with novel insights and improvements. Below, we address your queries:
>
> Initial Set: Indeed, the initial set originates from the intermediate phase of the standardized RANSAC process, notably the nearest neighbor with ratio test → SP iteration based on Equation (1) in the manuscript. The initial set can derive either from the ratio test result or the final SP outcome. Our experiments, as depicted in Table 1 in the main body of the paper, compare these two approaches. Notably, we present 'TP with SP' (utilizing the final SP result) and 'TP without SP' (employing the basic ratio test result). The intent was to gauge the impact of SP filtering on the hypothesis set. Interestingly, the integration of SP unexpectedly reduced the final accuracy. A plausible reason might be the incorrect exclusion of numerous prospective positive match pairs by SP.
>
> Saccade Function: Apologies for the oversight in elaboration. Given an initial matching pair (the hypothesis), the saccade function begins its traverse from this pair, covering the entire image region using the Breadth First Search (BFS). Simplistically, imagine it dividing the image into patches similar to the vision transformer. From the initial hypothesis matching patch pair, the saccade function employs BFS across all patches, selecting those that meet the three HR conditions. The actual trajectory is intriguing due to the patch location adjustment phase, resulting in patches of varied sizes and irregular locations compared to transformer's pre-segmented patches.
>
> Patch Location Adjustment: You've hit the mark; it indeed computes the smallest enclosing bounding box of the matched key points.
>
> On 'Hop 1' and 'Hop 2': We will certainly clarify this in the revised manuscript. To explain them, during image retrieval diffusion or propagation, it's unfeasible to diffuse across the entire database due to computational constraints. As a workaround, we limit the graph and diffuse among, say, the hop 3 neighbors of the query image, optimizing retrieval speed with minimal accuracy compromise. 'Hop 1' and 'Hop 2' denote propagation among hop 1 or hop 2 neighbors for each query, respectively. In our experiments, the hyperparameter was adjusted from hop 3 to hop 2 to curtail offline computational demands. To ensure fairness in comparison, we've included the hop 3 results in the rebuttal PDF. These further emphasize TP's superiority over SP.
>
> Once again, thank you for your invaluable feedback.
>
> Warm regards,
>
> Authors

---

### Official Review · Reviewer_X5kj · 2023-07-11

**Soundness:** 3 good
**Presentation:** 4 excellent
**Contribution:** 4 excellent
**Rating:** 7
**Confidence:** 4

**Summary:**

This paper proposes an approach in the area of image retrieval, more specifically landmark retrieval. Authors propose a new method for spatial verification that replaces standard and commonly used spatial model in RANSAC-based approaches, with topological one. Experiments show SOTA performance using handcrafted features, that even beats pretrained and is comparable to fine-tuned approaches.

**Strengths:**

+ Strong contributions: novel approach in hypothesis testing with RANSAC that is very different than traditional approaches and refreshing, inspiration from bio processes.
+ Very strong results with hand-crafted SIFT features and well established ranking framework with spatial verification.
+ Well written paper
+ Good experimental section, properly set baselines, good comparisons

**Weaknesses:**

- Time and memory analysis missing. It would be interesting how does this approach compares with standard RANSAC, and also how does the whole framework compare with competitive frameworks during inference
- This approach could be combined with other local features, even the trained or fine-tuned ones. It would make paper stronger if it was shown how well does it work with features beyond SIFT
- Spatial verification can be applied and help in visual geo-localizations on benchmarks such as Pittsburg30k and Tokyo24/7, it would be interesting to see how this method works there

**Questions:**

My first weakness is something that should be addressed in the rebuttal, ie the time and memory analysis. Other two weaknesses: ablate different features and visual geo-localization are improvements that would make paper stronger, but are not necessary.

**Limitations:**

Yes.

---

> ### Author Rebuttal · Authors · 2023-08-09
>
> Dear Reviewer X5kj,
>
> Thank you for your thorough feedback and for recognizing the strength of our contribution. We value your insights and would like to address each of your questions:
>
> Time and Memory Cost: We have provided details on time costs in lines 246-248 of our paper. Our TP, implemented in Python, averages 1.23s per image pair, making it slower than the C-implemented RANSAC SP at 0.53s. Nonetheless, transitioning to a C-based implementation promises substantial speed enhancements. The time distribution across the entire retrieval pipeline can be seen in Table 4 of the attached rebuttal pdf. When compared to the initial ranking stage (0.57s), the reranking stages of SP (53.2s) or TP (123.4s) appear relatively time-consuming. However, this disparity can be attributed to implementation techniques. Parallel implementation of SP/TP verification can drastically cut down time, particularly if the top 100 are verified concurrently. Regarding memory costs, these can be broken down into model size and local feature costs. Both SP and TP are lightweight and mobile-friendly, avoiding the cumbersome millions of parameters seen in models like ResNet. The memory cost for local features is identical for SP and TP. In scenarios where real-time feature calculation is feasible, this cost, such as with SIFT features, is minimized to an average of 0.43 MB per image (in numpy array format) in the Oxford set.  For tasks where memory consumption is a critical factor, the memory costs associated with our method can be mitigated. By leveraging real-time feature calculation, we can bypass storing extensive local features. This on-the-fly computation is about 0.16s per image pair.
>
>
> Compatibility with Other Local Features: We demonstrated the efficiency of integrating TP with DELG local features in paper table 2; our TP+DELG scored 86 and 71 on ROxford, surpassing the 85 and 59 of SP+DELG. In the attached rebuttal PDF, we also provide a comparison between SP+superpoint and TP+superpoint. The results show TP+superpoint scoring 69.5 and 46 on ROxford, as opposed to SP+superpoint's 66.1 and 42.4. Based on these comparisons across SIFT, DELG, and Superpoint, we posit that TP is a superior alternative to SP for rigid-body retrieval tasks.
>
> Geo-localization Experiments: At your suggestion, we conducted an evaluation on geo-localization. We used the renowned MSLS dataset. Our method's efficacy on the challenging MSLS validation set can be seen in table 3 of the attached pdf. Our approach notably outperforms SP when using SIFT features (top1 recall 0.59 vs. 0.74). Considering SP's prowess in geo-localization tasks, our TP's significant performance boost (25% improvement) was a pleasant surprise. After observation, we think the reason is that street views in geo-localization often deviate considerably from the plane assumption inherent in SP. This deviation means that the SP might primarily identify just one side or facet of the street, leading to its limitations in this scenario. In contrast, our TP demonstrates more adaptability under these conditions, making it a more versatile choice for such applications. Notably, our TP paired with SIFT even rivals the performance of the cutting-edge, fine-tuned R2Former model (CVPR '23), with top 1 recalls of 0.74 vs. 0.81, respectively. This experiment underscores our method's versatility.
>
> We are grateful for your thoughtful review and hope our clarifications prove satisfactory.
>
> Warm regards,
>
> Authors

---

### Author Rebuttal · Authors · 2023-08-09

Dear Reviewers,

First and foremost, we'd like to thank all the reviewers for the time and effort dedicated to reviewing our work. Your feedback has been instrumental in highlighting areas of improvement, and we've conducted additional experiments in response to your insightful comments.

Additional Experiments:

Table 1 offers a comparison encompassing more SIFT-based methods and those that are pre-trained, providing a broader landscape of performance benchmarks.

In Table 2, we delve into the performance of SP and our TP on the widely-recognized geo-localization benchmark MSLS.

Table 3 elucidates the results when SP and TP are amalgamated with DELG, specifically focusing on the hypergraph propagation among hop 3 neighbors.

Lastly, Table 4 provides a granular look at the time distribution throughout the retrieval process.

Your detailed feedback has not only enabled us to refine our work but has also guided our direction in conducting these supplementary experiments. We're optimistic that these additions provide a more comprehensive and robust foundation to our research.

Once again, thank you for your invaluable insights. We look forward to any further comments or suggestions you might have.

Warm regards,

Authors

---

> ### Comment · Reviewer_WpBp · 2023-08-13
> **Response to Authors' rebuttal**
>
> I appreciate the efforts made by the authors to prepare the rebuttal and the additional supporting experiments. It sufficiently addressed all my questions. After reading other reviews and the corresponding authors' responses, I hold my original opinion on this work and will keep my rating unchanged.

---

> > ### Author Response · Authors · 2023-08-21
> >
> > Thank you for recognizing our efforts in the rebuttal and experiments. We're glad we could address your questions. Your positive feedback and strong acceptance are greatly appreciated.

---

### Decision · Program_Chairs · 2023-09-21

**Decision:**

Accept (poster)

**Comment:**

Three reviewers recommended accepts (strong, weak, and borderline), while one reviewer gave a borderline reject. Most of the reviewers appreciated the novelty of the proposed bio-inspired topological verification that replaces traditional spatial verification for image retrieval, as well as solid experimental results, even with hand-crafted SIFT features. There were concerns about missing analyses and comparisons, but the authors successfully addressed most of them in the rebuttal. AC also finds that this paper has a good contribution to the community in the sense that it provides new insight and direction for the standard hypothesis testing process. thus recommending acceptance.